# Multiple Components of Protein Homeostasis Pathway Can Be Targeted to Produce Drug Synergies with VCP Inhibitors in Ovarian Cancer

**DOI:** 10.3390/cancers14122949

**Published:** 2022-06-15

**Authors:** Prabhakar Bastola, Gary S. Leiserowitz, Jeremy Chien

**Affiliations:** 1Department of Pharmacology, Toxicology & Therapeutics, University of Kansas Medical Center, Kansas City, KS 66160, USA; pbastola@askbio.com; 2Department of Obstetrics and Gynecology, University of California, Davis, CA 95817, USA; gsleiserowitz@ucdavis.edu; 3Department of Biochemistry and Molecular Medicine, University of California, Davis, CA 95817, USA

**Keywords:** VCP inhibitors, mifepristone, ovarian cancer, proteostasis, cancer therapy

## Abstract

**Simple Summary:**

The increased dependency of cancer cells on protein homeostasis creates a vulnerability that can be exploited using protein homeostasis inhibiting drugs. Recently, valosin-containing protein has emerged as a key component of the pathway and a novel therapeutic target for cancer treatment. The aim of our research is to explore the potential synergies between VCP inhibitors and other agents that disrupt protein homeostasis and to provide mechanistic understanding of drug synergies targeting multiple components of protein homeostasis. Our results indicate that VCP inhibitors can be combined with other endoplasmic reticulum stress-inducing agents to produce synergistic cytotoxicity in ovarian cancer cells.

**Abstract:**

Protein quality control mechanisms play an important role in cancer progression by providing adaptive responses and morphologic stability against genome-wide copy number alterations, aneuploidy, and conformation-altering somatic mutations. This dependency on protein quality control mechanisms creates a vulnerability that may be exploited for therapeutic benefits by targeting components of the protein quality control mechanism. Recently, valosin-containing protein (VCP), also known at p97 AAA-ATPase, has emerged as a druggable target in cancer cells to affect their dependency on protein quality control. Here, we show that VCP inhibitors induce cytotoxicity in several ovarian cancer cell lines and these compounds act synergistically with mifepristone, a drug previously shown to induce an atypical unfolded protein response. Although mifepristone at a clinically achievable dose induces a weak unfolded protein response, it enhances the cytotoxic effects of VCP inhibitor CB-5083. Mechanistically, mifepristone blocks the cytoprotective effect of ATF6 in response to endoplasmic reticulum (ER) stress while activating the cytotoxic effects of ATF4 and CHOP through the HRI (EIF2AK1)-mediated signal transduction pathway. In contrast, CB-5083 activates ATF4 and CHOP through the PERK (EIF2AK3)-mediated signaling pathway. This combination activates ATF4 and CHOP while blocking the adaptive response provided by ATF6, resulting in increased cytotoxic effects and synergistic drug interaction.

## 1. Introduction

Ovarian Cancer is the most lethal gynecologic malignancy, estimated to account for 12,080 deaths in 2022 in the United States. Currently, the standard treatment regimen includes surgical debulking and taxane-platinum combination chemotherapy. The combination treatment regimen shows a 70–80% initial response rate [1]. However, upon completion of chemotherapy, half of the patients with advanced disease will experience relapse within 18–22 months of remission [2]. Only a small fraction of patients with advanced disease (10–15%) achieve long-term remission with the standard chemotherapy [2]. The primary cause of relapse and eventual treatment failure is the preexisting intratumor heterogeneity and cellular phenotypic plasticity driven by genetic and epigenetic alterations [3]. The preexisting tumor heterogeneity and plasticity at the time of treatment afford some cancer cells with a proper adaptive response to chemotherapy to persist during treatment; these cells contribute to disease recurrence. In fact, the majority of patients who achieved a complete clinical response to the first-line chemotherapy will show residual disease on second-look laparotomy or will eventually relapse due to persistent disease [2].

After multiple rounds of chemotherapy, persistent tumor clones will expand and acquire additional genetic alterations that contribute to the acquired resistance. Therefore, it is critical to identify novel therapeutic targets and agents to treat this disease. Recently, several targeted therapies gained FDA approval in ovarian cancer, which includes bevacizumab (anti-vascular endothelial growth factor), and three PARP inhibitors: olaparib, rucaparib, and niraparib [4]. The clinical successes of these targeted therapies outline the relevance of developing novel therapeutic approaches in ovarian cancer. Several studies have attempted to identify novel therapeutic targets and pathways in ovarian cancer. Marcotte et al. performed a genome-wide screen of functional vulnerabilities with short hairpin RNAs (shRNAs) in 72 cancer cell lines, including 15 ovarian cancer cell lines [5]. The study identified genes in the ubiquitin proteasome system, including proteasome subunit alpha 1 (*PSMA1*) and proteasome subunit beta 2 (*PSMB2*), as essential genes in cancer cells. Similarly, Cheung et al. performed a separate genome-wide short hairpin RNA screening in 102 cell lines, including 25 ovarian cancer cell lines, which identified valosin-containing protein (*VCP/p97*), an important component of protein quality control, as one of the 22 putative genes essential in ovarian cancer cells [6]. Another study also identified VCP as one of the essential genes in cyclin E1 (CCNE1) amplified ovarian cancer cells [7].

Protein quality control includes adaptive pathways such as the unfolded protein response (UPR) and the chaperone activity of heat shock proteins (HSPs) that aid in the folding of misfolded and unfolded proteins. Additionally, protein quality control incorporates degradative pathways such as the ubiquitin proteasome system and autophagy that degrade unwanted or misfolded proteins into small peptides or individual amino acids [8]. Oncogenic insults such as increased mutational burden, copy number alterations, chromosomal duplications/deletions, and oncogene-induced oxidative stress manifest a higher burden on the protein quality control mechanisms, making cancer cells more reliant on these mechanisms for survival and proliferation, a phenomenon known as non-oncogenic addiction [9]. Because cancer cells are addicted to or highly reliant on protein quality control mechanisms, targeting these pathways would be beneficial in cancer therapeutics. This idea led to the development of agents that target elements of protein quality control mechanisms for cancer therapy, such as bortezomib and carfilzomib [10,11]. Targeting protein quality control mechanisms can be extended further in designing effective drug combinations. Both the restorative and destructive components of the protein quality control pathways could be selectively modulated to elevate the cytotoxic effect mediated by a single agent. An example of targeting the restorative component of the protein quality control pathway includes the use of heat-shock protein 90 (Hsp90) inhibitors. Interestingly, Hsp90 serves as a capacitor of phenotypic variations and morphologic evolution, determines the adaptive properties of genetic variations [12,13], and therefore could potentially affect the evolution of normal and cancer genomes. Thus, targeting components of protein quality control in cancer cells may be viewed not only as an approach to target non-oncogene addiction but also as a novel way to limit the adaptability and evolution of cancer genomes. In this paper, we set out to explore the possibility of targeting multiple components of the protein quality control pathways with relevant drug candidates.

Previously, we have shown that quinazoline-based VCP inhibitors, such as DBeQ and ML240, produce dose-dependent cytotoxicity [14]. Although these compounds are specific and potent VCP inhibitors, they lacked proper pharmacological properties, exhibiting low solubility and a lack of bioavailability [15]. Using the pharmacophore from ML240, Anderson et al. identified an oral VCP inhibitor, CB-5083 [15]. The study reported enhanced biochemical inhibition towards VCP, increased in vitro cytotoxicity in several cancer types, as well as pronounced in vivo efficacy in multiple mouse tumor xenograft models with CB-5083 [15]. The results prompted the initiation of two first-in-class Phase I clinical trials of CB-5083 in hematological cancers and solid tumors. However, both clinical trials were terminated due to ocular side effects caused by the off-target inhibition of PDE6 [16]. Therefore, it would be important to identify synergistic combinations in which CB-5083 can be used at a lower concentration to limit its ocular side effects while maintaining the cytotoxic effects in cancer cells. In our current study, we investigated the in vitro efficacy of CB-5083 in ovarian cancer and identified new strategies to enhance the cytotoxic effect of VCP inhibitors in combination with other compounds that modulate the unfolded protein response.

To enhance the cytotoxic effect of VCP inhibitors, we decided to focus on compounds that modulate the unfolded protein response. Several inhibitors have been identified over the years that regulate specific branches of the unfolded protein response [8,17,18,19]. However, none of these compounds have gained FDA approval. Mifepristone (RU-486) is an FDA-approved oral progesterone receptor antagonist, and it has been used in the clinic for ending early term pregnancy. Over the years, several additional targets for mifepristone have been identified, including the glucocorticoid receptor [20] and the nuclear receptor subfamily 1 [21]. Multiple studies have focused on the anti-cancer effect of mifepristone in meningioma [22], triple negative breast cancer [23], and ovarian cancer [24]. Additionally, several studies have now reported that a clinically achievable dose of mifepristone induces the unfolded protein response [24,25,26]. However, the mechanism resulting in the induction of the unfolded protein response is unknown. Here, we show that a clinically achievable dose of mifepristone is synergistic with VCP inhibition in ovarian cancer cells. Furthermore, we report that mifepristone treatment inhibits the activating transcription factor 6 (ATF6) branch of the unfolded protein response and induces the expression of activating transcription factor 4 (ATF4) through activation of the heme-regulated inhibitor (HRI) kinase pathway. Our results identify a plausible mechanism of mifepristone-induced endoplasmic reticulum (ER) stress response and establish a relevant drug combination based on targeting protein quality control mechanisms.

## 2. Materials and Methods

### 2.1. Reagents

CB-5083 (S8101, Selleckchem, Houston, TX, USA), NMS-873 (S7285, Selleckchem), DBeQ (SML0031, Sigma-Aldrich, Saint Louis, MO, USA), STF-083010 (S7771, Selleckchem,), STF-083010 (SML0409, Sigma-Aldrich), ISRIB (SML0843, Sigma-Aldrich) and mifepristone (M8046, Sigma-Aldrich) were dissolved in dimethyl sulfoxide (DMSO) at 50 mM stock solution. Tunicamycin (T7765, Sigma-Aldrich) was dissolved in DMSO at 10 mM stock solution and ISRIB (SML0843, Sigma-Aldrich) was dissolved in DMSO at 5 mM stock solution. All stock solutions were aliquoted in small volumes and stored at −80 °C. Before use, appropriate concentrations of these compounds were prepared by dissolving the stock solution (or its subsequent dilution) into the appropriate growth media.

### 2.2. Cell Lines and Cell Culture

Cell lines OVCAR10, OVCAR5, SKOV3 and RMG1 were cultured in Medium 199 (M5017, Sigma-Aldrich,) and MCDB 105 (M6395, Sigma-Aldrich,) at 1:1 ratio with 5% fetal bovine serum (F0926, Sigma-Aldrich,) and 1% streptomycin/pencillin (PSL01, Caisson Labs, Smithfield, UT, USA). Cell lines OVSAHO, OVCAR8 and IGROV1 were cultured in RPMI (RPL03, Caisson Labs) with 5% fetal bovine serum and 1% streptomycin/penicillin. PERK-MEF^−/−^ (CRL-2976, ATCC, Manassas, VA, USA) and GCN2-MEF^−/−^ (CRL-2978, ATCC) were cultured in DMEM, 0.1 mM non-essential amino acids (MT25025CL, Fisher, Waltham, MA, USA), 0.05 nM 2-mercaptoethanol (M3148, Sigma-Aldrich), 10% fetal bovine serum, and 1% streptomycin/penicillin. MDA-MB-241 cells were cultured in RPMI, 0.1 mM non-essential amino acid with 5% fetal bovine serum and 1% streptomycin/penicillin. All cell lines used in this study were cultured in a 37 °C humidified incubator with 5% CO_2_ and were periodically checked for mycoplasma contamination. MEF cells were purchased directly from the vendor for this study. The identity of all remaining cell lines was confirmed with STR genotyping.

### 2.3. Cell Viability Assay and Drug Synergy Studies

Briefly, 5000 cells/well were plated in a 96-well clear plate and were allowed to incubate overnight in a humidified incubator at 37 °C with 5% CO_2_. The next day, cells were treated with the vehicle or compounds of interest by replacing the existing media. Plates were then allowed to incubate for 72 h, and cell growth was assessed using the sulforhodamine B assay protocol as previously described [14].

Drug synergy was determined by calculating the combination indexes (CIs) by dividing the expected effect by the observed effect [14]. The expected values assume additive effects between two drugs. N represents the total number of combination indexes (CIs) determined from 16 different drug combinations in duplicates that produced 20–80% of cytotoxic effect from three independent experiments. For combination studies, CB-5083 doses ranged between 0.1 μM and 1 μM and mifepristone doses ranged between 5 μM and 20 μM with varying dose ratios.

### 2.4. Clonogenic Assay and 3D Spheroid Assay

For the clonogenic assay, 1000 cells/plate were plated as single cell suspension in a 6-well plate and were allowed to incubate overnight in a 37 °C humidified incubator with 5% CO_2_. The next day, cells were plated with vehicle, single agent, or a combination by replacing the existing media with media containing these agents. Cells were then allowed to incubate for 48 h in the 37 °C humidified incubator with 5% CO_2_. Media containing the agents were then replaced with regular media and cells were subsequently allowed to grow for an additional 8–10 days, replacing the media every other day.

Optimal colonies were stained with sulforhodamine B staining dye for 30 min. Excess dye was aspirated and washed with 1% acetic acid. Plates were allowed to air-dry, and colonies were then photographed using the ChemiDoc MP Imaging System (Bio-Rad, Hercules, CA, USA). Colonies were counted using Quantity One (version 4.6.9) (Bio-Rad). Colony numbers were subsequently plotted as percent growth using Prism (ver. 7) (Graphpad, San Diego, CA, USA).

For the 3D spheroid assay, 25,000 cells were plated in each well of a 96-well ultra-low attachment plate (4520; Corning, Corning, NY, USA) and allowed to incubate at a 37 °C humidified incubator with 5% CO_2_ for 4 days to form spheroids. At day 4, spheroids were treated with vehicle, CB-5083, and/or mifepristone. Treatment was performed for 72 h at which point cell viability was determine using the 3D CellTiter-Glo Viability Assay (G9681; Promega, Madison, WI, USA) according to the manufacturer’s protocol.

### 2.5. Transient Small Interfering RNA (siRNA) Knockdown and Plasmid Transfection

All targeting siRNA oligos were selected using IDT’s predesigned siRNA selection tool (HRI siRNA = hs.Ri.EIF2AK1.13.2 and PKR siRNA = hs.Ri.EIF2AK2.13.2). Based on the design tool, the selected siRNAs showed no off-targeted binding. Scrambled negative control DsiRNA (51-01-19-09; IDT, Coralville, IA, USA) was purchased from IDT. All siRNAs were dissolved in a 20 µM stock solution in nuclease-free duplex buffer (IDT, 11-01-03-01). Transient siRNA transfections were performed using oligofectamine (12252-011; Life Technologies, Carlsbad, CA, USA) according to the manufacturer’s protocol. Briefly, 0.5 × 10^6^ cells/well were plated in a 6-well plate with appropriate media without antibiotics. The next day, 3 µL of oligofectamine reagent was mixed with 12 µL of Opti-MEM media, and 10 µL of 20 µM siRNA stock oligos were mixed with 175 µL of Opti-MEM media. Both mixtures were allowed to incubate at room temperature for 5 min. The diluted oligos were then combined with the diluted oligofectamine solution and the final mixtures were allowed to incubate at room temperature for 15 min. In the meantime, media in the cell lines were replaced with pre-warmed Opti-MEM media. The final siRNA oligo mixtures were added dropwise, and the cells were placed back in the humidified incubator. After 4 h, Opti-MEM media with serum was added without replacing the transfection mixture. Cells were collected at the indicated times and equal proteins were immunoblotted to check for the transfection efficiencies.

### 2.6. Caspase-3 Activity Assay

DEVD-Afc (ab285386, Abcam, Cambridge, UK) was purchased from Abcam. Caspase-3 activity assay was performed according to our previously published protocol [14]. Briefly, 20 µg of total protein was combined with 2 µL of 2 mM DEVD-Afc in 96-well flat-bottom plates (3296; Corning). Caspase buffer was added to the wells to make the final volume 200 µL/well. Plates were then incubated at 37 °C. After 2 h, fluorescence measurements were taken using a plate reader at an excitation wavelength of 400 nm and an emission wavelength of 510 nm.

### 2.7. Western Blot and Antibodies

For all comparative western blots (except for siRNA experiments), 0.5 × 10^6^ cells/well were plated in a 6-well plate with appropriate growth media and cells were allowed to incubate overnight. The following day, cells were treated with media containing appropriate concentrations of the vehicle (DMSO) or the different compounds used in the study for indicated time. Cells were then collected, washed with PBS and lysed with 100 µL of 2× Laemmli Buffer with 5% 2-Mercaptoethanol at 95 °C for 15 min.

Gel electrophoresis was performed by loading equal volumes of the protein lysates onto SDS-PAGE and transferred on PVDF/nitrocellulose membrane. After the transfer, membranes were blocked with 3% BSA solution made in TBS-T for 1 h. Membranes were then incubated with primary antibodies diluted in 3% BSA solution at 1:1000 dilution mostly overnight at 4 °C. Membranes were subsequently washed 3 times in TBS-T solution and incubated with respective secondary HRP conjugated antibodies diluted in 3% BSA solution at 1:5000 dilution for 1 h at room temperature. Lastly, membranes were washed 3 times in TBS-T and developed with the Thermo Femto (34096, Fisher) or Thermo Dura (37071, Fisher) reagents using the Bio-Rad Imager. All uncropped western blot from the main figures can be found in the Appendix A.

The primary antibodies, purchased from Cell Signaling Technology (Danvers, MA, USA), included PARP (#9542), total caspase-3 (#9665), cleaved caspase-3 (#9661), Grp78 (#3177), CHOP (#5554), β-actin (#3700), ubiquitin (#3933), ATF6 (#65880), PERK (#3192), ATF4 (#11815), IRE1α (#3294), XBP1 (#12782), GCN2 (#3302) and PKR (#12297), except p-eIF2α (ab32157, Abcam) and p-IRE1α (nb100-2323, Novus, Centennial, CO, USA). Secondary antibodies, including HRP-linked anti-rabbit (#7074) and HRP-linked anti-mouse (#7076), were purchased from Cell Signaling Technology.

## 3. Results

### 3.1. VCP Inhibitor CB-5083 Treatment Induces Cytotoxicity in Ovarian Cancer Cells

We started off by analyzing the genetic dependence of over 1000 pan-cancer cell lines towards VCP based on the CRISPR essentiality screen data in the Cancer Dependency Map [27]. The results demonstrated VCP as a common essential gene across the cancer cell lines with strong genetic dependency towards ovarian cancer cell lines (Appendix A). Consistent with this observation, we previously observed the dose-dependent cytotoxic effect of VCP inhibitors DBeQ and ML240 in a panel of ovarian cancer cell lines [14]. However, the cytotoxic effect of the oral VCP inhibitor CB-5083 in these cells was unknown. Therefore, we performed a sulforhodamine B (SRB) assay to analyze cell viability following the treatment with incremental doses of CB-5083 up to 25 μM for 72 h in high-grade serous ovarian cancer cell lines OVCAR10, OVCAR8, OVSAHO, and OVCAR5 as well as in clear cell ovarian cancer cell lines RMG1 and SKOV3. Consistent with other compounds in its class, CB-5083 treatment showed a dose-dependent cytotoxicity (Figure 1A) with half-maximal growth inhibition (GI_50_) ranging from 0.46 ± 0.07 μM to 0.94 ± 0.23 μM (Figure 1B). These GI_50_ values are comparable to previously reported half-maximal growth inhibition values in the lung carcinoma cell line A459, the colon carcinoma cell line HCT116 [15,28], and patient-derived organoid models for ovarian cancer [29]. Our results suggest that CB-5083 can effectively inhibit in vitro cell growth in high-grade serous and clear cell ovarian cancer.

### 3.2. VCP Inhibitors Show Synergistic Cytotoxicity with Clinically Achievable Doses of Mifepristone

Treatment with VCP inhibitors results in the induction of unfolded protein response-mediated apoptosis [14,15]. Hence, we attempted to identify compounds that could produce synergistic cytotoxic effects with VCP inhibitors so that these compounds could be considered as potential clinical candidates for combination therapy with VCP inhibitors. Previously, we reported that inhibition of growth arrest and DNA damage-inducible 34 (GADD34) by salubrinal results in synergistic cytotoxicity with VCP inhibitors, including CB-5083, in several ovarian cancer cell lines [14]. Although the demonstration of synergy between VCP inhibitors and salubrinal provided proof of concept that multiple agents inhibiting the protein quality control pathway can be combined to achieve synergy; the clinical relevance is not prominent because salubrinal is not a clinical candidate. Several studies have now indicated that treatment with mifepristone (RU-486), an anti-progesterone receptor inhibitor, results in the induction of the unfolded protein response [25,26]; however, the molecular mechanism contributing to the unfolded protein response by mifepristone is not well characterized. Given that mifepristone is an FDA-approved drug, we decided to test the potential synergistic cytotoxicity between CB-5083 and mifepristone and to investigate the molecular mechanisms contributing to a potential synergy.

We performed sulforhodamine B (SRB) assays in the panel of ovarian cancer cell lines tested in Figure 1 with clinically achievable concentrations of CB-5083 (0.1–1 μM) and mifepristone (5–20 μM). Our results indicate that most cell lines show a synergistic effect between these two compounds (Figure 2A and Appendix A). The synergistic effects were also observed when CB-5083 was substituted with other VCP inhibitors DBeQ and NMS-873 suggesting that mifepristone enhances the cytotoxic effect of VCP inhibitors in ovarian cancer cells (Figure 2B,C and Appendix A). To further corroborate our synergistic studies, we performed colony formation assays in RMG1 and OVSAHO cells treated with vehicle (DMSO), CB-5083, mifepristone, or the combination of CB-5083 and mifepristone. Our results indicate that in RMG1 cells, single-agent treatment with 0.5 μM CB-5083 or 20 μM mifepristone only shows a modest reduction in colony formation, while the combined treatment significantly suppresses colony formation (Figure 2D). Similar results were observed with 0.75 μM CB-5083 and 20 μM mifepristone (Figure 2D) as well as in another ovarian cancer cell line OVSAHO (Figure 2E). The combination of DBeQ and mifepristone also show synergistic activity at higher concentrations of DBeQ in the OVSAHO ovarian cancer cell line (Appendix A). Additionally, synergistic cytotoxicity between CB-5083 and mifepristone was observed in an RMG1 spheroid model (Appendix A). RMG1 cells showed the strongest synergistic effects among the cell lines we tested; hence, we decided to use this cell line in subsequent studies to further investigate the molecular mechanisms contributing to the synergistic effect produced by CB-5083 and mifepristone.

### 3.3. CB-5083 and Mifepristone Combination Enhances Caspase Activity and Cytotoxicity

To understand the potential mechanism of cell death with CB-5083 and mifepristone, we treated RMG1 with vehicle (DMSO), single-agent CB-5083, mifepristone, or the combination of CB-5083 and mifepristone for 12 h or 24 h. We observed a reduction in full-length PARP and total caspase-3 starting at 12 h as well as a robust induction of cleaved caspase-3 at 24 h (Figure 3A). Similarly, caspase activity assays showed a threefold increase in activity at 18 h with the combination of CB-5083 and mifepristone (Figure 3C). These results indicate that the combination is significantly more effective in inducing caspase-mediated cell death than a single agent. Given that both CB-5083 and mifepristone have been shown to induce the unfolded protein response [14,15,25,30,31], we checked two prominent unfolded protein response markers, namely glucose-regulated protein 78 (Grp78/HSPA5), C/EBP homologous protein (CHOP/DDIT3), and total K48 linked poly-ubiquitinated proteins. We observed a robust induction of both Grp78 and CHOP in cells treated with the combination or CB-5083 alone but not in cells treated with mifepristone alone (Figure 3A). Consistent with previous studies [15], we observed an increase in poly-ubiquitinated proteins following CB-5083 treatment (Figure 3B). The combination also increases K48 ubiquitination of proteins (Figure 3B), indicative of a block in proteasome-mediated proteolysis. Similar results are observed in another cell line, OVSAHO, treated with the combination (Appendix A). At the tested concentration, mifepristone by itself did not affect the levels of K48-ubiquitinated proteins but attenuated the levels of poly-ubiquitinated proteins induced by CB-5083.

### 3.4. Mifepristone Activates the Unfolded Protein Response Independent of Glucocorticoid, Estrogen, and Progesterone Receptor Inhibition

Mifepristone (RU-486) was developed as an anti-progesterone inhibitor, and was later shown to be a potent inhibitor of glucocorticoid receptors [32]. To investigate the potential role of progesterone and glucocorticoid receptors toward the mechanism of synergistic cytotoxicity, we used two different cell lines, one lacking the progesterone receptor (MDA-MB-468) [33] and another lacking the glucocorticoid receptor (IGROV1) [34]. We treated MDA-MB-468 cells with 10–40 µM mifepristone, CB-5083 (1 µM), or a combination of CB-5083 (1 µM) and mifepristone (20 µM). We observed induction of ATF4 and CHOP by mifepristone at higher concentrations, indicating that these cells can induce mifepristone-mediated unfolded protein response (Figure 4A). Moreover, we observed synergistic cytotoxicity between mifepristone and CB-5083 in these cells (Figure 4B). Similarly, glucocorticoid receptor-negative IGROV1 ovarian cancer cells displayed enhanced cytotoxicity to DBeQ and mifepristone (Figure 4C). These results indicate that the unfolded protein response mediated through mifepristone is independent of glucocorticoid and progesterone receptor activity.

### 3.5. Mifepristone Blocks ATF6 Signaling

Previous studies have indicated that mifepristone induces unfolded protein response [26]; however, the mechanism underlying this response has not been fully understood. To further investigate the mechanism, we first incubated RMG1 cells with a clinically achievable dose of mifepristone (20 μM) and harvested cells at different time-points between 1 and 24 h. Similarly, we harvested cells incubated with different doses of mifepristone (5–80 μM) for 18 h. Subsequently, whole cell lysates were tested with antibodies against all three branches of the unfolded protein response. 20 μM mifepristone treatment did not show any difference in full-length ATF6 (ATF6-FL) (Figure 5A,B). High doses of mifepristone (40–80 μM) decrease the expression of full-length ATF6; however, this may be due to cell death observed at these high doses when treated for 18 h (Figure 5B). This was confirmed by conducting cell viability assays in RMG1 to calculate half-maximal growth inhibition (GI_50_) upon mifepristone treatment. GI_50_ values for mifepristone were 26.4 μM and 27.9 μM (Appendix A).

Next, with 20 μM mifepristone treatment, we observed the induction of ATF4 at 24 h (Figure 5A). The induction of ATF4 was observed starting at 20 μM followed by slight induction of CHOP at 20 μM (Figure 5A,B) followed by robust induction at 40 μM and 80 μM (Figure 5B). We also observed a robust inhibition of p-IRE1α and induction of total IRE1α with increasing dose of mifepristone (Figure 5B). In these experiments, we were unable to detect the induction of Grp78 by mifepristone (Figure 5A,B). The induction of Grp78 serves as an adaptive response to aid in the folding of misfolded and unfolded protein in the endoplasmic reticulum, allowing cells to resolve the unfolded protein response. Therefore, we reasoned that mifepristone may be exerting an inhibitory effect on IRE1 or ATF6 branch (or both), which would inhibit the expression of Grp78 and attenuate the adaptive response when it is combined with VCP inhibitors, thereby resulting in enhanced cytotoxicity.

To define the potential effect of mifepristone on IRE1α signaling, we treated the cells with tunicamycin, an N-linked glycosylation inhibitor, to induce ER stress and activate the unfolded protein response in the presence of varying mifepristone. As a control, we performed a similar experiment with STF-083010 (IRE1α inhibitor). Tunicamycin treatment activates unfolded protein response as evidenced by the increased expression of spliced XBP1 (sXPB1) and Grp78 (Figure 5C, the last lane). Interestingly, although mifepristone at lower concentrations (≤10 µM) does not block Grp78, it attenuates the induction of Grp78 at concentrations above 20 µM (Figure 5C). Our results indicate that both mifepristone and STF-83010 caused a dose-dependent attenuation of Grp78 (Figure 5C and Appendix A). Although increasing doses of STF-083010 caused a dose-dependent attenuation of spliced XBP1 (XBP1s), the expression of spliced XBP1 did not change with mifepristone treatment (Figure 5C and Appendix A). Therefore, mifepristone may inhibit the IREα signaling downstream of XBP1s.

Next, we determined the effect of mifepristone on CB-5083 induced ER stress and the unfolded protein response. We treated RMG1 cells with increasing doses of CB-5083 (between 1 and 5 μM) for 6 h. We observed an increase in spliced XBP1 (s-XBP1) expression starting at 2.5 μM of CB-5083 (Figure 5D). Once again, we did not observe the attenuation of spliced XBP1 (s-XBP1) with the addition of 20 μM mifepristone (Figure 5D). These results indicate that mifepristone does not block the endonuclease activity of IRE1 kinase. In addition to the IRE1 branch, the ATF6 branch also induces Grp78 expression through the binding of nuclear ATF6 (ATF6-N) to the ER stress-responsive element (ERSE) [35]. To investigate the potential effect of mifepristone on the ATF6 branch, we determined ATF6 expression following the mifepristone and CB-5083 treatment. We saw cleavage of full-length ATF6 (ATF6-FL) with 2.5 μM and 5 μM of CB-5083; however, this cleavage was blocked with the addition of 20 μM mifepristone (Figure 5D). These results suggest that mifepristone inhibits the adaptive IREα signaling downstream of XBP1s and ATF6 signaling by blocking its proteolytic activation.

### 3.6. Mifepristone Activates Heme-Regulated Inhibitor (HRI) Pathway to Induce Activating Transcription Factor 4 (ATF4)

Activating transcription factor 4 (ATF4) is activated upon phosphorylation of eukaryotic initiation factor 2 alpha (eIF2α), which can be phosphorylated by four upstream kinases: PERK, GCN2, HRI, and PKR (please see the list of abbreviations for full forms). To investigate the potential involvement of PERK and GCN2 in ATF4 activation, we incubated mouse embryonic fibroblast generated from PERK knockout mice (PERK-KO-DR) and mouse embryonic fibroblast generated from GCN2 knockout mice (GCN2-KO-DR) with 20 μM of mifepristone and harvested whole cell lysates at 6 and 12 h. Similarly, we treated both mouse embryonic fibroblasts (MEFs) with 2 μM tunicamycin for 10 h, which acted as positive controls for the experiment. As expected, 2 μM tunicamycin was unable to induce the expression of ATF4 in PERK-KO-DR; however, in both conditions ATF4 expression was induced with 20 μM mifepristone (Figure 6A). These results indicate that mifepristone activates ATF4 independent of PERK or GCN kinases. Next, we analyzed the effect of kinases HRI and PKR towards the activation of ATF4. We transiently transfected small interfering RNAs (siRNAs) targeting HRI and PKR for 48 h and followed with 20 μM mifepristone treatment for 6 and 10 h. Our results indicate that siRNA targeting HRI was able to attenuate ATF4 expression (Figure 6B). Taken together, these results suggest that ATF4 activation upon mifepristone treatment is mediated through the HRI pathway.

### 3.7. Targeting Multiple Components of Proteostasis Produces Synergistic Drug Interactions

To further discern if modulation of the unfolded protein response could result in synergistic cytotoxicity when combined with VCP inhibitor CB-5083, we treated RMG1 cells with IRE1α inhibitor STF-083010 (STF), integrated stress response pathway inhibitor (ISRIB) and ATF6 pathway inhibitor S1P inhibitor in combination with CB-5083. The results, shown in Figure 7A, indicate that STF reduces the number surviving cells at or above 10 µM. However, at these higher concentrations of STF, we did not observe drug synergism with CB-5083. In contrast, we observed drug synergism between CB-5083 in combination with ISRIB or the S1P inhibitor (Figure 7B,C). These results suggest that modulation of these two pathways by mifepristone may contribute to drug synergism between CB-5083 and mifepristone. Given that we saw drug synergism between CB-5083 and ISRIB, we confirmed this synergism between CB-5083 and ISRIB by clonogenic assay. Results, shown in Figure 7D, indicate that CB-5083 and ISRIB show synergistic interactions at the highest tested doses. Moreover, the triple combination of CB-5083, mifepristone, and ISRIB produced the greatest synergy (Figure 7E). Similar results were observed in another ovarian cancer cell line OVSAHO (Appendix A). These results suggest that targeting multiple components of the protein homeostasis pathway represents a promising therapeutic strategy to targeting cancer cells.

## 4. Discussion

The endoplasmic reticulum (ER) consists of interconnected tubules and flattened sacs that extend through the entire cytoplasm. The primary functions of the endoplasmic reticulum include post-translational modifications of nascent proteins, folding of secretory proteins, synthesis of lipids, and storage of calcium. Several physiological insults such as calcium dysregulation, glucose or nutrient deprivation, increased reactive oxygen species, and increased burden of misfolded and unfolded proteins in the endoplasmic reticulum can trigger the unfolded protein response. The unfolded protein response is primarily an adaptive response that is manifested through phosphorylation of two transmembrane endoplasmic reticulum kinases, IRE1 and PERK, and cleavage of endoplasmic reticulum transmembrane protein ATF6. Over the years, several studies have shown that the unfolded protein response can be targeted in multiple human disorders, including cancer. Previously, we showed that the unfolded protein response pathway can be induced in ovarian cancer by inhibiting VCP [14]. VCP inhibitors induce an unrestrained unfolded protein response that eventually initiates caspase-mediated cell death. Furthermore, we showed that VCP inhibitors can be combined with salubrinal potentiating the effect of VCP inhibitors [14]. In this study, our primary objective was to identify a clinically relevant agent that could enhance the cytotoxic effect of VCP inhibitors by modulating the unfolded protein response in ovarian cancer cells. Several compounds that modulate several branches of the unfolded protein response have been identified over the years; however, none have been FDA-approved. Through this study, we identified strong synergistic cytotoxicity between mifepristone and several classes of VCP inhibitors at a clinically achievable dose of mifepristone in ovarian cancer cells. Given that VCP has also been shown to be an essential gene in CCNE1 amplified ovarian cancer cells [7], future studies could analyze if CB-5083 and mifepristone combination display enhanced synergistic cytotoxicity in CCNE1 amplified ovarian cancer cells. Consistent with this idea, the analysis of VCP knockout effects in ovarian cancer cells from the Dependency Map project [27,36] indicates CCNE1-amplified cells, such as COV318, ONCODG1, OVCAR3, and SNU8, show strong dependency on VCP (Appendix A). Additionally, since oral VCP inhibitors display a pan-cancer cytotoxicity, our results have broad implications in terms of designing future combination clinical trials with VCP inhibitors. Recently, OVCAR5 has been shown to be gastrointestinal in origin [37]. Our results in OVCAR5 provides further support towards exploring this synergistic cytotoxicity in a pan-cancer context.

Mifepristone was initially approved as an anti-progesterone receptor inhibitor [38]; however, subsequent studies have identified other targets of mifepristone including several reports suggesting that mifepristone treatment displays an atypical unfolded protein response [26]. We decided to investigate the mechanism of this atypical unfolded protein response upon mifepristone treatment. Mifepristone treatment resulted in the inhibition of Grp78 expression (an endoplasmic reticulum chaperone) as well as the induction of ATF4. Although we observed a decrease in phospho-IRE1α and an increase in total IRE1α expression with mifepristone treatment, the endoribonuclease activity of IRE1α was not affected by mifepristone. Next, we analyzed the changes in the ATF6 branch. Our results indicate that unlike tunicamycin and CB-5083, mifepristone inhibits the cleavage of full length ATF6 (ATF6-FL) (Figure 5C,D).

Recently, Gallagher et al. used endoplasmic reticulum stress-responsive element (ERSE) to screen compounds that could inhibit ATF6 and showed that Ceapins, a class of pyrazole amides, could selectively inhibit ATF6α. The study showed that Ceapsin-A7 (ATF6α inhibitor) enhanced the cytotoxicity of thapsigargin (unfolded protein response inducer) [39]. Results from our synergy studies between VCP inhibitors (unfolded protein response inducer) and mifepristone (ATF6 branch inhibitor) are therefore in agreement with their observation. However, further studies are needed to confirm that the synergistic effect between mifepristone and VCP inhibitors is mediated via this mechanism.

Through this study, we found that mifepristone treatment activates the heme-regulated inhibitor (HRI) pathway, resulting in ATF4 activation. Induction of ATF4 increases the expression of CHOP and enhanced CHOP expression has been shown to induce the unfolded protein response-mediated cell death. This underlines a parallel mechanism of action that could result in a potential synergy between mifepristone and VCP inhibitors (Figure 8). It is possible that both the activation of the HRI pathway and inhibition of the ATF6 branch confers the final synergistic cytotoxicity in our experiments. Subsequent studies need to be performed to analyze the effect of each of these targets to conclusively identify the mechanism of cytotoxicity mediated through mifepristone. It is interesting to note that the gene expression analysis of high-grade serous carcinomas from The Cancer Genome Atlas datasets, available from UALCAN [40], cbioportal [41], and KM plotter [42], indicates relatively higher expression of HRI (EIF2AK1) compared to PERK (EIF2AK3) or PKR (EIF2AK2) and higher expression of pro-survival gene GRP78 compared to pro-apoptotic gene CHOP (Appendix A). However, genetic alterations in these UPR-related genes are infrequent (Appendix A), but expression of CHOP, GRP78, and HRI are associated with overall survival of patients with high-grade serous carcinoma (Appendix A).

We also observed synergistic effects between CB-5083 and mifepristone in IGROV1 (shown to have low GR expression) and triple-negative breast cancer (TNBC) cell line-MDA-MB-468. These results suggest that the atypical unfolded protein response via mifepristone is independent of glucocorticoid receptor, estrogen receptor, and progesterone receptor inhibition; however, we are unable to rule out a potential role of mifepristone through unknown steroids or growth factors inhibition. In this study, we identified exciting and clinically relevant candidates that can be used in the combination chemotherapy. Similarly, we discovered a previously uncharacterized mechanism of mifepristone in modulating the unfolded protein response. Mifepristone has been shown to be a safe compound even when given over a long interval. We envision mifepristone to be used as a maintenance therapy together with other modulators of the protein quality control pathways such as VCP inhibitors, heat-shock protein 90 (Hsp90) inhibitors, and autophagy modulators.

The rationale for a combination therapy is to use drugs that work by different mechanisms and can display increased effects when used together. Chemotherapeutic agents are often associated with drug-related toxicities. Identifying compounds that are synergistic allows lowering the dose of these agents, which could lower the drug-related toxicities. Relevant to this study is the ocular side effect of CB-5083 that resulted in a premature termination of a Phase I clinical trial. The results of the trial and clinical doses that caused ocular side effects have not been published. With the eventual publication of the study, we would be able to determine if synergistic combinations that we observed with mifepristone might allow the evaluation of this combination in a clinical trial.

Previously, we have shown that treatment with CB-5083 could result in the development of resistance in ovarian cancer cells [8,30], therefore targeting tumor cells with a combination of compounds could avoid the development of drug resistance. Our results have implications not just for ovarian cancer therapy, but results from our studies are relevant for designing combination therapies for other tumors. Since VCP inhibitors display cytotoxicity towards multiple cancer types, the combination of VCP inhibitors and mifepristone should show efficacy in other cancer types.

Similarly, we demonstrated that cytotoxicity mediated through VCP inhibitors can be enhanced by modulating the unfolded protein response. With scores of compounds that modulate the unfolded protein response in the drug development pipeline, our studies provide a rationale for combining the compounds that induce the terminal unfolded protein response with the compounds that inhibit the ubiquitin proteasome system. Follow-up studies should focus on assessing the potential synergistic cytotoxicity between VCP inhibitors and compounds that have been shown to modulate the unfolded protein response, considering that CB-5083 clinical trial was stopped due to ocular side effect [16]. These synergy studies may allow for the use of a lower dose of CB-5083 to minimize ocular toxicity while enhancing cytotoxic effect in tumor cells. In the future, high-throughput drug screening can be performed to identify possible FDA-approved compounds that display synergistic cytotoxicity with VCP inhibitors. Identifying FDA-approved drug candidates that are synergistic with VCP inhibitors will enhance the efficacy of VCP inhibitors as well as negate the possible occurrence of drug resistance and ocular side effects.

## 5. Conclusions

Our results show that VCP inhibitors can be combined with other ER stress inducers or UPR modulators to produce synergistic cytotoxicity in ovarian cancer cells. Our results have implications not just for ovarian cancer therapy but are relevant for designing combination therapies for other tumors. Since VCP inhibitors display cytotoxicity towards multiple different cancer types, the combination of VCP inhibitors and mifepristone may show efficacy in other cancer types.

## Figures and Tables

**Figure 1 cancers-14-02949-f001:**
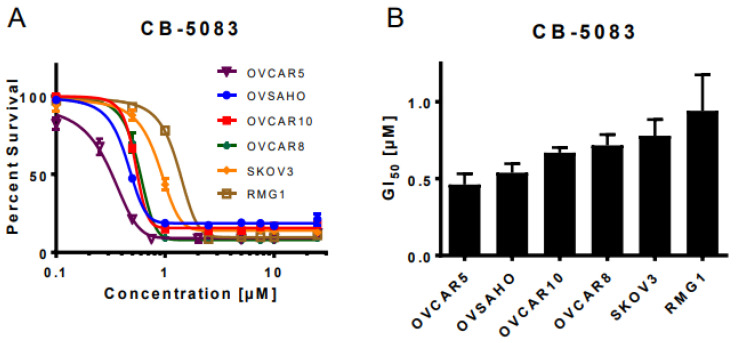
CB-5083 treatment is cytotoxic in ovarian cancer cell lines. (**A**): Ovarian cancer cell lines, namely OVCAR5, OVSAHO, OVCAR10, OVCAR8, SKOV3, and RMG1 were treated with increasing doses of CB-5083 ranging from 0.1 µM to 25 µM for 72 h. Dose–response curves were generated using GraphPad Prism based on the four parameters of nonlinear regression. The curves were constrained at the top (100%) and the bottom (>0%). Every point in the dose–response curve represents mean ± SEM taken from three technical replicates for all cell lines. (**B**): The bar graph represents mean GI_50_ + SEM taken from four biological replicates in all cell lines.

**Figure 2 cancers-14-02949-f002:**
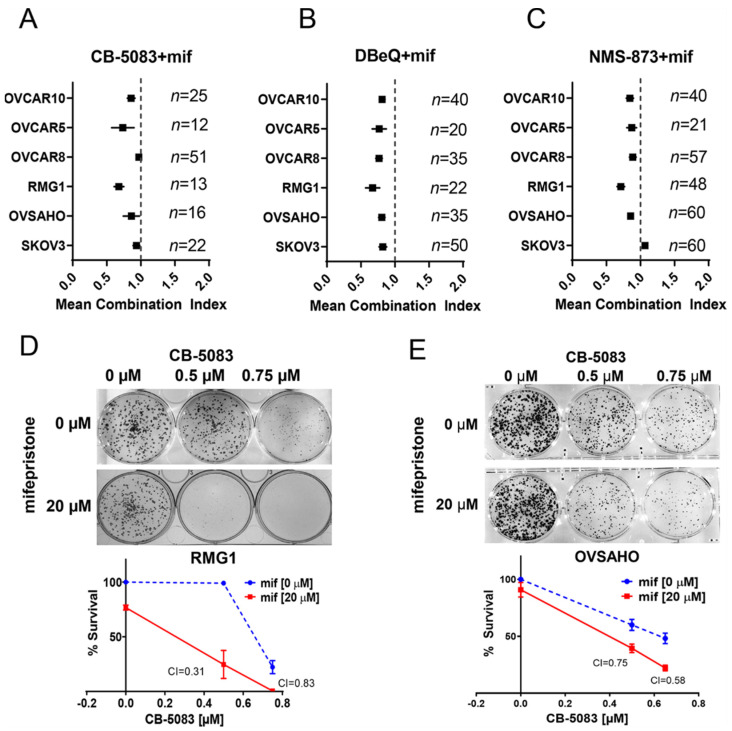
Mifepristone produces synergistic cytotoxicity with VCP inhibitors. (**A**): High-grade serous ovarian cancer cell lines OVCAR10, OVCAR5, OVCAR8, and OVSAHO as well as clear cell ovarian cancer cell lines RMG1 and SKOV3 were treated with different combinations of mifepristone (between 5 and 20 µM) and CB-5083 (between 0.25 and 1 µM). Combination indexes (CIs) were calculated at all such combinations that yielded an effect between 20 and 80% cytotoxicity. Each data point is shown as the mean CI of all calculated combinations and 95% confidence interval. “*n*” represents the total number of CIs calculated from the combinations that produced the combined effect of 20–80% cytotoxicity from three biological replicates. Mean combination index (CI) less than 1 indicates synergy. (**B**): Same as (**A**) but treated with different combinations of mifepristone (between 5 and 20 µM) and DBeQ (between 1 and 7.5 µM). (**C**): Same as (**A**) but treated with different combinations of mifepristone (between 5 and 20 µM) and NMS-873 (between 1 and 2.5 µM). (**D**): RMG1 cells were treated with the indicated concentrations of CB-5083 and mifepristone for 48 h followed by 6–8 days of recovery in regular medium. Mean percent survival ± SEM was calculated based on the number of colonies from three biological replicates. (**E**): Same as (**D**) but in OVSAHO cells. Percent survival was calculated based on the number of colonies from three biological replicates.

**Figure 3 cancers-14-02949-f003:**
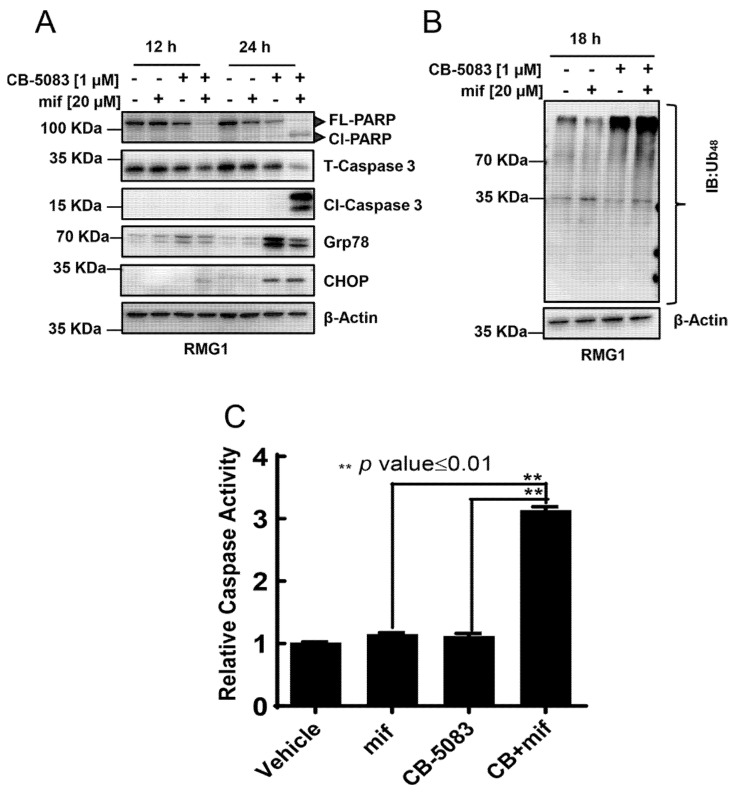
Caspase activation. (**A**): Western blot analysis of total (T) and cleaved (Cl) caspase-3, full length (FL) and cleaved PARP1, and ER stress-related proteins at 12 h and 24 h after CB-5083 and/or mifepristone (mif). (**B**). Immunoblot (IB) analysis of ubiquitinated (K48-linked) proteins following CB-5083 or mifepristone treatment. Both (**A**,**B**) were performed in at least two biological replicates. Uncropped immunoblots can be found in Appendix A. (**C**). Analysis of caspase activities following indicated treatment at 18 h. *p* values were calculated using the two-tailed Student’s *t*-test.

**Figure 4 cancers-14-02949-f004:**
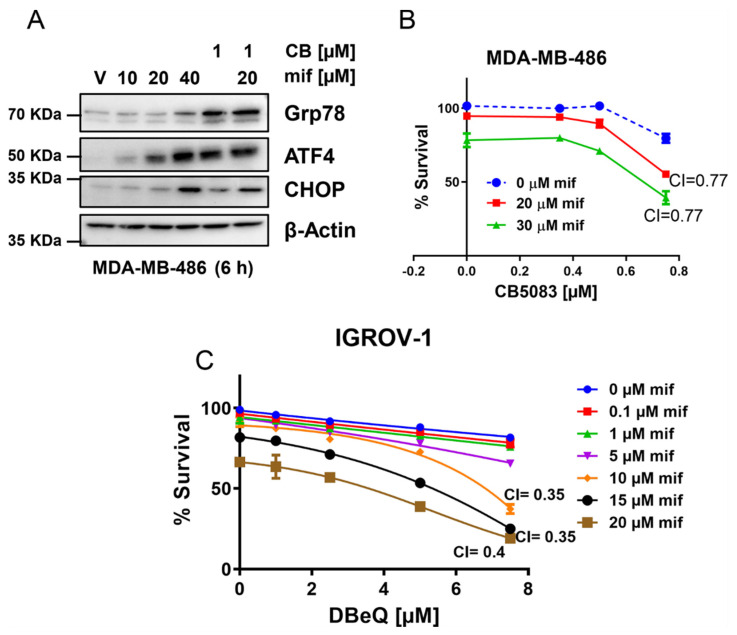
Mifepristone activates the unfolded protein response independent of glucocorticoid, estrogen and progesterone receptor inhibition. (**A**): Triple-negative breast cancer cell line (MDA-MB-468) was incubated with indicated compounds for 6 h. Whole cell lysates were then subjected to immunoblotting with antibodies for indicated proteins. β-Actin was used as loading control. Uncropped immunoblots can be found in Appendix A. (**B**): MDA-MB-468 cells were treated with different concentrations of CB-5083 up to 0.75 μM and different concentrations of mifepristone (20 μM and 30 μM) for 72 h. Cell viability was measured using the SRB assay and dose–response curves were plotted with GraphPad Prism. CI stands for combination index. (**C**): IGROV1 cells were treated with indicated concentrations of DBeQ and mifepristone for 72 h. Cell viability was measured using SRB assay and dose–response curves were generated using GraphPad Prism. All experiments were performed in three biological replicates.

**Figure 5 cancers-14-02949-f005:**
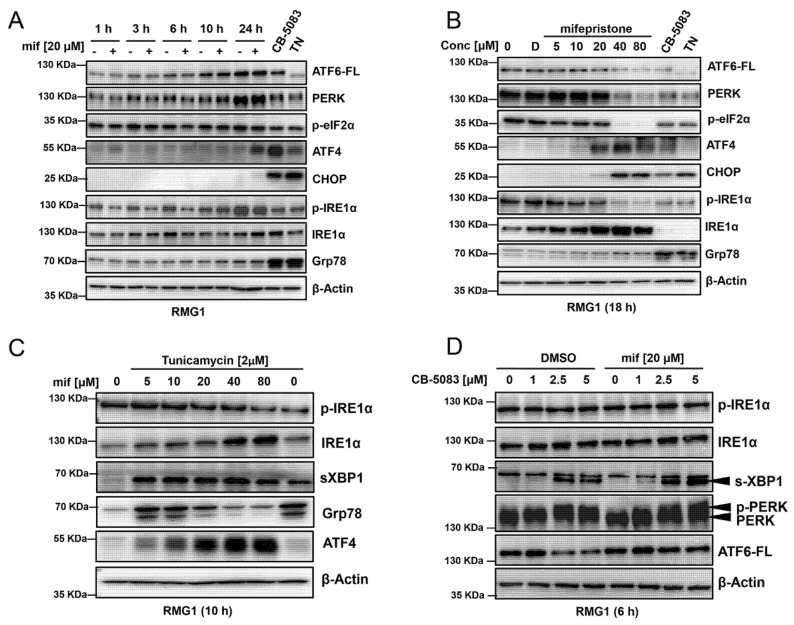
Mifepristone shows atypical modulation of the unfolded protein response. (**A**): RMG1 cells were incubated with vehicle (DMSO) or 20 μM of mifepristone and samples were harvested at 1, 3, 6, 10 and 24 h. Whole cell lysates were subjected to immunoblotting and probed with the antibodies for several proteins in the unfolded protein response pathway. 2.5 μM of CB-5083 and 2 µM of TN (Tunicamycin) treated for 24 h were analyzed as positive controls. β-Actin was used as a loading control. (**B**): RMG1 cells were incubated with media (0), DMS0 (D), different doses of mifepristone (between 5 μM and 80 μM), 2.5 μM of CB-5083 and 2 μM of tunicamycin (TN). All samples were incubated for 18 h, and protein lysates were immunoblotted with several antibodies for proteins in the unfolded protein response pathway. β-Actin was used as loading control. (**C**): RMG1 cells were incubated with increasing doses of mifepristone (5–80 µM) with 2 μM of tunicamycin for 10 h. Cells were harvested, and proteins were subjected to immunoblotting to analyze the indicated proteins. (**D**): RMG1 cells were incubated with increasing doses of CB-5083 (1–5 µM) with or without mifepristone (20 µM) for 6 h. Cells were harvested, and proteins were subjected to immunoblotting to analyze the indicated proteins. β-Actin was used as a loading control. All experiments were performed in at least two biological replicates. Uncropped immunoblots can be found in Appendix A.

**Figure 6 cancers-14-02949-f006:**
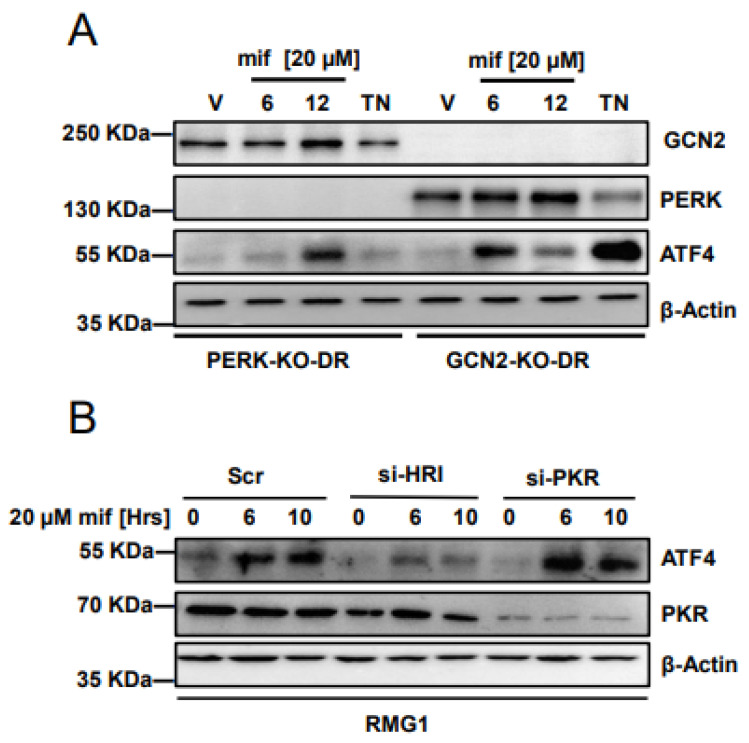
Mifepristone activates the HRI kinase pathway. (**A**): Mouse embryonic fibroblast generated from PERK knockout mice (PERK-KO-DR) and mouse embryonic fibroblast generated from GCN2 knockout mice (GCN2-KO-DR) were incubated with 20 µM of mifepristone for 6 and 12 h. Vehicle (V) and TN (2 μM-Tunicamycin) treatments were performed for 12 h and lysates were probed with indicated antibodies. β-actin was used as loading control. (**B**): RMG1 cells were transiently transfected with scrambled (Scr) and small interfering RNAs (siRNAs) targeting HRI (si-HRI) and PKR (si-PKR) for 48 h followed by treatment of 20 µM of mifepristone for 6 and 10 h. Protein lysates were then subjected to immunoblotting with indicated antibodies. β-actin was used as loading control. All experiments were performed in at least two biological replicates. Uncropped immunoblots can be found in Appendix A.

**Figure 7 cancers-14-02949-f007:**
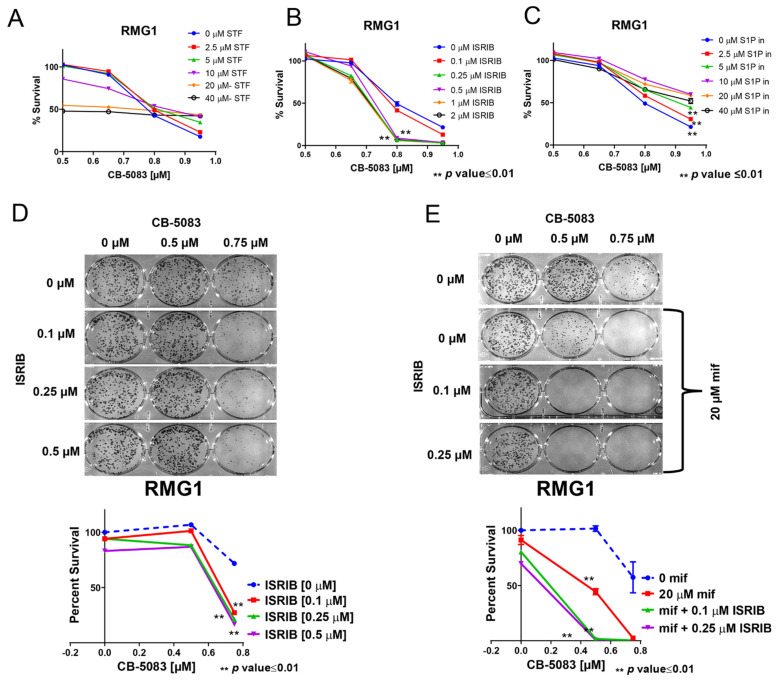
Synergistic drug interactions between CB-5083 and specific ER stress inducers. Dose response to the combination of CB-5083 and an IREα inhibitor (STF) (**A**), integrated stress response pathway inhibitor (ISRIB) (**B**), or ATF6 pathway inhibitor S1P (**C**) in RMG1 cells. (**D**): Clonogenic assay to assess the combined effect of CB-5083 and ISRIB. (**E**): Clonogenic assay to assess the triple combination of CB-5083, ISRIB, and mifepristone. Mean percent survival ± SEM was calculated based on the number of colonies from three biological replicates. *p* values were calculated in reference to CB-5083 treatment alone using the two-tailed Student’s *t*-test.

**Figure 8 cancers-14-02949-f008:**
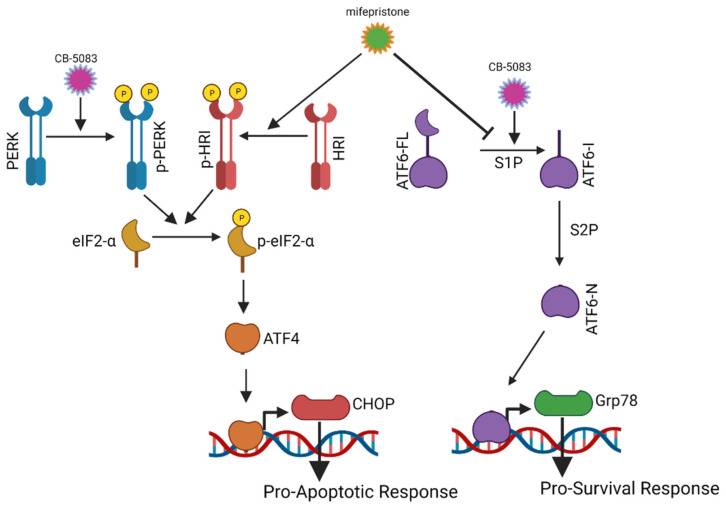
Putative mechanism for synergistic cytotoxicity between CB-5083 and mifepristone. Treatment with VCP inhibitor-CB-5083 activates the activating transcription factor 6 (ATF6) branch and the protein kinase R-like endoplasmic reticulum kinase (PERK) branch. Activation of ATF6 can result in the induction of glucose-regulated protein 78 (Grp78). Induction of Grp78 is considered a pro-survival response. Similarly, PERK activation can ultimately induce CCAAT/enhancer-binding protein homologous protein (CHOP). Enhanced expression of CHOP can be a pro-apoptotic response. Mifepristone treatment inhibits the ATF6 branch and enhances CHOP expression through the activation of the heme-regulated inhibitor (HRI) kinase pathway. The combination of these two effects could result in the enhanced cytotoxicity seen with CB-5083 and mifepristone combination.

## Data Availability

The data presented in this study is available in this article (and Appendix A).

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
