# Peer review of "Multiple Components of Protein Homeostasis Pathway Can Be Targeted to Produce Drug Synergies with VCP Inhibitors in Ovarian Cancer"

_cancers, 2022, doi:10.3390/cancers14122949_

Round 1

Reviewer 1 Report

The manuscript is interesting and well written. It require some suggestions to improve the quality of this study. Bastola et al, identified that VCP 19 inhibitors can be combined with other endoplasmic reticulum stress-inducing agents to produce 20 synergistic cytotoxicity in ovarian cancer cells. There are some suggestions that can improve the manuscript for publication.

Suggestions:

  1. Fig.1- There is no normal ovarian surface epithelial cells as a control to compare. I would recommend to see the cytotoxic effect of CB-5083 in cancer cells as compared to normal OSEs.
  2. Fig, 7 D-E; Please check the font size in the quantitative data. Its overlap each other.
  3. Please also check the status of HRI kinase pathway related genes in cancer datasets through cBIOPORTAL, UALCAN, software.
  4. Method section: 3D spheroid formation assay- There is a lack of in vivo data. I highly recommend to the authors provide 3-D spheroids data. The author may take an idea of 3-D spheroid formation assay from the Cell Reports and Cancer Research- latest papers where authors show the 3-D spheroid formation assay Protocol in a better way. Parashar D et al - Cell Reports 2019 (PMID: 31875548), Geethadevi et al 2021,- Cancer Research (PMID: 34380633).
  5. Please also perform some invitro assay such as migration and Invasion assay, Cell cycle arrest, and  Apoptosis assay by FACS to show the impact of synergetic effect.

Author Response

Reviewer#1

  1. Fig.1- There is no normal ovarian surface epithelial cells as a control to compare. I would recommend to see the cytotoxic effect of CB-5083 in cancer cells as compared to normal OSEs.

Response: We would like to thank the reviewer for the suggestion to include the effect of CB-5083 treatment in “normal cells”. While this study is important to show efficacy of CB-5083 in ovarian cancer cells, given the shorten timeline to submit this revision, we currently do not have the capability to conduct the suggested study. Additionally, CB-5083 has already been shown to be orally bioavailable with no major side effects based on in vivo studies (PMID# 26555175, 28878026). The GI50 values from our in vitro studies are similar to the published GI50 in A459 and HCT116 (PMID# 26555175, 28878026). Previous publications have shown xenograft tumor reduction in A459, HCT116, and patient-derived xenografts with minimal side effects (PMID# 26555175). These lines of evidence suggest a selective cytotoxicity of CB-5083 toward cancer cells. Finally, considering that drug treatment often is preceded by debulking surgery, the effect of investigational drugs on normal ovarian surface epithelial cells may not be as relevant as their effects on other normal cells.  

  1. Fig, 7 D-E; Please check the font size in the quantitative data. Its overlap each other.

Response: Thank you. We have corrected the font conversion problems that existed in some pdf figures, including figures 7 D-E.

  1. Please also check the status of HRI kinase pathway related genes in cancer datasets through cBIOPORTAL, UALCAN, software.

Response: Thank you for the suggestion. We analyzed UALCAN web portal for the UPR pathway related genes including the HRI kinase pathway. The analysis of gene expression in the high-grade serous carcinoma dataset indicates that HRI gene expression (in transcript per million counts) is higher than two other related genes, PERK and PKR. For the downstream genes involved in the pathway, pro-apoptotic genes, such as CHOP, is lower than pro-survival genes, such as GRP78. Expression of three genes (CHOP, GRP78, and HRI) are associated with overall survival in patients with high-grade serous carcinomas. These results are included as Supplementary Figure S14. (Line 568-575)

  1. Method section: 3D spheroid formation assay- There is a lack of in vivo data. I highly recommend to the authors provide 3-D spheroids data. The author may take an idea of 3-D spheroid formation assay from the Cell Reports and Cancer Research- latest papers where authors show the 3-D spheroid formation assay Protocol in a better way. Parashar D et al - Cell Reports 2019 (PMID: 31875548), Geethadevi et al 2021,- Cancer Research (PMID: 34380633).

Response: We agree with the reviewer that 3D spheroid formation assay would help to examine the combination treatment more robustly. Therefore, we have included the results from our 3D formation assay between CB-5083 and mifepristone in Supplementary S3C (Line 304-305). In addition, we included a reference to our prior publication testing the efficacy of CB-5083 in several patient-derived organoid models, including cancer cells from a heavily pre-treated platinum resistant patient (Chen et al, PMID: 32253045) (Line 260-261).

  1. Please also perform some invitro assay such as migration and Invasion assay, Cell cycle arrest, and  Apoptosis assay by FACS to show the impact of synergetic effect.

Response: We would like to thank the reviewer for suggesting we perform further in vitro assays such as migration assay, invasion assay, cell cycle arrest assay and apoptosis assay by FACS to show the impact of synergistic effect. In this manuscript, we focused our studies in identifying the molecular mechanism of the synergy between CB-5083 and mifepristone. We agree with the reviewer that a comprehensive set of in vitro studies could be of interest to the ovarian cancer therapeutics community; however, we suggest these assays should be part of separate future study.  

Reviewer 2 Report

In this study, Bastola et al have attempted to identify new combination strategies that can improve outcomes in ovarian cancer patients, which is associated with poor overall survival in patients. The authors used various ovarian cancer cell lines that are representative of high-grade serous ovarian carcinoma and clear cell carcinoma for their studies. They identified that VCP inhibitors in combination with mifepristone exhibits synergistic cytotoxic effects in their model systems and further the synergistic effects were independent of mifepristone’s effects on other steroid receptors. They were able to show that these effects of mifepristone were mediated by the HRI-mediated signal transduction pathway and VCP inhibitor mediated its cytotoxic effects via PERK. The study is novel and provides a mechanism-based rationale for using this drug combination to achieve synergistic effects in ovarian cancer; however, there are some concerns that are listed below:

  • Introduction section:
    • The introduction section is verbose and needs shortening. For instance, there is no need to summarize information of clinical trials that lead to the approval of targeted therapies for ovarian cancer by FDA. Just a sentence describing the current FDA approved drugs would suffice.
    • Lines 116-117, please specify what ‘lack of proper pharmacological properties’ mean. Are there issues related to pharmacokinetics or bioavailability of drugs or toxic effects?
    • Lines 124-126, the authors state that combination therapy is rational approach to lower ocular side effects. Please include information as to beyond what dose/ plasma concentration levels these effects are observed. Please address this aspect in the discussion section of the manuscript as well as to how the concentration ranges tested in these in vitro combination experiments are lower than those that show toxic effects.

  • Materials & Methods section:
    • OVCAR-5 cell line has been recently categorized as originating from the gastrointestinal tract and not of ovarian origin (PMID: 27353327). Please address this in the limitations of the study in the discussion section of the manuscript.
    • Please also elaborate on the CCNE mutation status of these cell lines and did that have an effect on the sensitivity of the various cell lines to the VCP inhibitor? Please comment.
    • In the combination index experiment, please provide a range of cell numbers that were used for the different cell lines in the experiment.
    • Please include the reference for the method used to determine combination index.
    • For the combination index experiments, were a fixed or varying ratio for the doses/ concentrations of the two drugs used? Please clarify e.g. 1:1 ratio of two drugs or 2:1 ratio etc.
    • Given the off-target effects reported with siRNAs, it seems unclear as to how many siRNA constructs were tested by the authors and what were their sequences? Also, please provide information about the % knockdown that was achieved from using siRNA.
    • Please mention the dilutions used in western blotting for various primary antibodies.

  • The authors should also test the effects of the drugs shown here on normal ovarian surface epithelial cells (like HOSE cell line) and/ or fallopian tube epithelial cells (FTE cell lines) to illustrate that the drugs are specific for the ovarian cancer cells and do not damage the normal/ healthy cells.
  • Just like the introduction, the discussion section is also extremely long and the authors seem to digress from the important conclusion of the paper. The authors are suggested to make the discussion more succinct and relevant to the observations/ results obtained.
  • It will be interesting to see whether these drugs indeed show synergy in the preclinical animal models of ovarian cancer given the kinetics of the drugs change dramatically in a physiological system relative to an in vitro setting. Preclinical study data would strongly support the observations from the paper.

Author Response

Reviewer#2

  • Introduction section:
    • The introduction section is verbose and needs shortening. For instance, there is no need to summarize information of clinical trials that lead to the approval of targeted therapies for ovarian cancer by FDA. Just a sentence describing the current FDA approved drugs would suffice.

Response: We agree with the reviewer and have summarized the indicated portion in the introduction.

    • Lines 116-117, please specify what ‘lack of proper pharmacological properties’ mean. Are there issues related to pharmacokinetics or bioavailability of drugs or toxic effects?

Response: We have updated this section to include the information requested by the reviewer.

    • Lines 124-126, the authors state that combination therapy is rational approach to lower ocular side effects. Please include information as to beyond what dose/ plasma concentration levels these effects are observed. Please address this aspect in the discussion section of the manuscript as well as to how the concentration ranges tested in these in vitro combination experiments are lower than those that show toxic effects.

Response: We thank the reviewer for the comment. We revised the discussion to include the implication of current studies to address the clinical limitation of CB-5083 as it relates to ocular side effect (Line 603-607).

  • Materials & Methods section:
  • OVCAR-5 cell line has been recently categorized as originating from the gastrointestinal tract and not of ovarian origin (PMID: 27353327). Please address this in the limitations of the study in the discussion section of the manuscript.

Response: We thank the reviewer for bringing this to our attention. Our results indicate synergistic cytotoxicity in OVCAR5 suggesting a broader implication of this combination strategy in a pan-cancer context. We have included this as part of our discussion (Line 538-540).

    • Please also elaborate on the CCNE mutation status of these cell lines and did that have an effect on the sensitivity of the various cell lines to the VCP inhibitor? Please comment.

Response: Thank you for this excellent suggestion. We have included this in our discussion: Line 530-535).

    • In the combination index experiment, please provide a range of cell numbers that were used for the different cell lines in the experiment.

Response: Thank you for asking for this clarification. We have added the suggested information in Line 164.

    • Please include the reference for the method used to determine combination index.

Response: Thank you for pointing that out. We have added the suggested information in Line 169-170.

    • For the combination index experiments, were a fixed or varying ratio for the doses/ concentrations of the two drugs used? Please clarify e.g. 1:1 ratio of two drugs or 2:1 ratio etc.

Response: Thank you for allowing us to clarify this section. We have added the suggested information in Lines 173-175.

    • Given the off-target effects reported with siRNAs, it seems unclear as to how many siRNA constructs were tested by the authors and what were their sequences? Also, please provide information about the % knockdown that was achieved from using siRNA.

Response: Thank you for the suggestion. We have updated the section to include this information.

    • Please mention the dilutions used in western blotting for various primary antibodies.

Response: Thank you for pointing that out. We have added the suggested information in Line 231.

  • The authors should also test the effects of the drugs shown here on normal ovarian surface epithelial cells (like HOSE cell line) and/ or fallopian tube epithelial cells (FTE cell lines) to illustrate that the drugs are specific for the ovarian cancer cells and do not damage the normal/ healthy cells.

Response: We would like to thank the reviewer for the suggestion to include the effect of CB-5083 treatment in “normal cells”. While this study is important to show efficacy of CB-5083 in ovarian cancer cells, given the shorten timeline to submit this revision, we currently do not have the capability to conduct the suggested study. Additionally, CB-5083 has already been shown to be oral bioavailable with no major side effects in vivo studies. Our GI50 values from our in vitro studies seem to be quite similar to the published GI50 in A459 and HCT116 (PMID# 26555175, 28878026). Previous publications have shown tumor reduction in A459 and HCT116 in vivo with minimal side effects. Based on these lines of evidence, we strongly believe that CB-5083 will show enhanced toxicity towards ovarian cancer cells compared to “normal ovarian surface epithelial cells”. 

  • Just like the introduction, the discussion section is also extremely long and the authors seem to digress from the important conclusion of the paper. The authors are suggested to make the discussion more succinct and relevant to the observations/ results obtained.

Response: We agree with the reviewer and have summarized the discussion.

  • It will be interesting to see whether these drugs indeed show synergy in the preclinical animal models of ovarian cancer given the kinetics of the drugs change dramatically in a physiological system relative to an in vitro setting. Preclinical study data would strongly support the observations from the paper.

Response: We agree with the reviewer that a preclinical study would indeed be extremely valuable to further evaluate the synergistic effect observed in this study; however, due to time constraint for revision, we would not be able to address this issue at this time. However, we will address this issue in future studies.

Round 2

Reviewer 2 Report

The authors have addressed all the comments and the manuscript is much improved